# CURVATURE-CORRECTED LEARNING DYNAMICS IN DEEP LINEAR NEURAL NETWORKS

## ABSTRACT

Deep neural networks exhibit complex learning dynamics due to highly non-convex loss landscape. Second order approaches, such as natural gradient descent, mitigate such problems by neutralizing the effect of potentially ill-conditioned curvature, yet it is largely unknown how the current theory of deep learning generalizes beyond gradient descent to these higher order learning rules. To answer these questions, we derive exact solutions to learning dynamics of deep linear networks under a spectrum of curvature-corrected learning rules. Our analysis reveals that curvature corrected learning preserves a core feature of gradient descent, a conservation law, such that the learning trajectory follows precisely the same path in the underlying manifold as gradient descent, only accelerating the temporal dynamics along the path. We also show that layer-restricted approximations of natural gradient, which are widely used in most second order methods (*e.g.* K-FAC), can significantly distort the learning trajectory into highly diverging dynamics that significantly differs from true natural gradient, which may lead to undesirable network properties. We also introduce fractional natural gradient that applies partial curvature correction, and show that it provides most of the benefit of full curvature correction in terms of convergence speed, with additional benefit of superior numerical stability and neutralizing vanishing/exploding gradient problems, which holds true also in layer-restricted approximations.

## 1 INTRODUCTION

Difficulty in training deep neural networks arises from the fact that the network's input-output map $f_\theta(\cdot)$ is nonlinearly related to its parameters $\theta$. This causes non-convex loss landscape with proliferation of saddle-points and poorly-conditioned curvature where gradient-based first order optimization methods perform poorly (Martens, 2010; Dauphin et al., 2014). Second order methods, such as natural gradient descent (Amari, 1998), compensate for the effect of curvature by using the distance metric intrinsic to the space of input-output maps to define the update steps (Pascanu & Bengio, 2013; Martens, 2014; Bernacchia et al., 2018), rather than the parameter space. Recent advancements led to approximate implementations of these methods that prove efficient for practical scale applications (Ba et al., 2016; Grosse & Martens, 2016; Martens et al., 2018; Osawa et al., 2019).

Despite their practical effectiveness, however, the exact nature of such curvature-corrected learning process remains largely unknown. Do curvature-corrected learning methods simply accelerate convergences towards the *same* minimum solutions as gradient descent, or do they impose implicit bias toward qualitatively *different* solutions?

As a first step toward establishing theoretical understanding of these questions, we analyze the exact learning dynamics of deep linear networks under a spectrum of curvature-corrected update rules. Deep linear networks provide an excellent mathematical framework for developing insightful theoretical understanding of the complex inner workings of deep nonlinear networks (Goodfellow et al., 2016). Despite their simplicity, deep linear networks capture the essential nonlinear relationship between network's input-output maps and their parameters, and exhibit comparable learning behavior to their nonlinear counterparts that can be exactly solved for rigorous analysis. Indeed, many recent works analyzed the learning trajectories of deep linear networks under gradient descent to compute the convergence rate under various initial conditions (Arora et al., 2018a;b; Bartlett et al., 2019; Du & Hu, 2019), revealed decoupled modes of convergence dynamics to explain the origin of multiple

stage-like loss profiles (Saxe et al., 2013), and showed the implicit bias for regularization (Du et al., 2018; Arora et al., 2019) and resistance to overfitting (Advani & Saxe, 2017; Lampinen & Ganguli, 2018; Poggio et al., 2018). Yet, it is uncertain whether these convergence properties generally apply for update rules beyond gradient descent.

**Our contribution** The main results are summarized as follows.

1. We derive a generalized conservation law that describes the optimization paths of network parameters under gradient descent as well as curvature-corrected update rules. Consequently, curvature correction only affects the speed of convergence without affecting other qualitative properties of parameter update process.

2. There is a trade-off between map dynamics and parameter dynamics. The full curvature correction effect of natural gradient descent (NGD) completely linearizes the map learning dynamics of deep networks, equivalent to that of shallow networks. Such complete linearization, however, sacrifices stability of parameter update dynamics to explode when gradient vanishes and vice versa.

3. We introduce a regularized version of NGD that partially corrects for the effect of curvature, called $\sqrt{\text{NGD}}$, which facilitates the parameter update dynamics by eliminating the vanishing/exploding update problems. This makes the map dynamics slightly nonlinear, but no more so than that of single hidden layer networks under gradient descent.

4. NGD makes the learning process prone to overfitting by simultaneously learning both the signal and the noise dimensions of data, whereas $\sqrt{\text{NGD}}$ partially retains gradient descent's resistance to overfitting by separating the time-scales between the signal and the noise dimensions.

5. The widely-used block-diagonal approximation of NGD breaches the aforementioned conservation law, resulting in highly divergent parameter update dynamics, which breaks the weight balance across layers. In contrast, block-diagonalization of $\sqrt{\text{NGD}}$ preserves stability of parameter update dynamics, yielding efficient and stable learning algorithms.

## 2 SETUP AND NOTATIONS

Consider a depth $d$ network that consists of an input layer, $d - 1$ hidden layers, an output layer, and weight matrices $\boldsymbol{w} \equiv \{w_i\}_{i=1}^d$ that connect the adjacent layers. The network's input-output map is $\bar{w} \equiv \prod_{i=1}^d w_i = w_d \cdots w_1$, such that $f_{\boldsymbol{w}}(x) = \bar{w}x$. The network learns the input-output statistics of a dataset $D = \{x^\mu, y^\mu\}_{\mu=1}^P$ by minimizing the squared-error loss:

$$L(\boldsymbol{w}) = \frac{1}{2}\mathbb{E}_D[\|\bar{w}x - y\|^2] = \text{Tr}\left[\frac{1}{2}(\bar{w} - \bar{w}_*)\Sigma_x(\bar{w} - \bar{w}_*)^\intercal\right] + const,$$

where $\mathbb{E}_D$ is the expectation over the dataset $D$, $\Sigma_x \equiv \mathbb{E}_D[xx^\intercal]$ is the input correlations, and $\bar{w}_* \equiv \mathbb{E}_D[yx^\intercal]\Sigma_x^{-1}$. Neglecting the constant term, the loss function is expressed as

$$L(\boldsymbol{w}) = \text{Tr}\left[\frac{1}{2}\Delta\Sigma_x\Delta^\intercal\right]. \qquad (\Delta \equiv \bar{w} - \bar{w}_*) \qquad (1)$$

where $\Delta$ denotes the displacement between $\bar{w}$ and $\bar{w}_*$.

Shallow networks ($d = 1$, $\bar{w} = w_1$) exhibit linear learning dynamics under gradient descent, whose convergence rates scale with eigenvalues of $\Sigma_x$. In this case, curvature correction has the well-understood effect of normalizing the convergence rates, which is also achievable by simple pre-whitening of input correlation. Instead, we are interested in the less-understood effect of how curvature correction facilitates the complex nonlinear dynamics of deep networks ($d \geq 2$). Therefore, we consider pre-whitened input distribution $\Sigma_x = I$ to isolate the nonlinear effect of curvature correction, but this condition is not critical for the analysis.

**Gradient and Hessian$_+$** We use bold symbols to collectively represent network parameters and derivatives in array form. For example, $\dot{\boldsymbol{w}} \equiv \begin{bmatrix} \dot{w}_1 \\ \dot{w}_2 \end{bmatrix}$ and $\boldsymbol{g} \equiv \begin{bmatrix} \frac{\partial L}{\partial w_1} \\ \frac{\partial L}{\partial w_2} \end{bmatrix} = \begin{bmatrix} w_2^\mathsf{T} \Delta \\ \Delta w_1^\mathsf{T} \end{bmatrix}$ represent the continuous-time weight update and the gradient of a depth $d = 2$ network. Hessian is fully characterized by its operation on weight update, which, by definition, produces gradient update:

$$\boldsymbol{H}\dot{\boldsymbol{w}} = \dot{\boldsymbol{g}} = \begin{bmatrix} w_2^\mathsf{T}\dot{\Delta} + \dot{w}_2^\mathsf{T}\Delta \\ \dot{\Delta} w_1^\mathsf{T} + \Delta \dot{w}_1^\mathsf{T} \end{bmatrix}. \qquad (\dot{\Delta} = \dot{\bar{w}} = w_2\dot{w}_1 + \dot{w}_2 w_1) \qquad (2)$$

However, true Hessian-based methods (*e.g.* Newton-Raphson method) can converge to any extrema types. To guarantee convergence to (local) minimum solutions, natural gradient methods use positive semi-definite (PSD) approximations of Hessian (*e.g.* Fisher matrix (Amari, 1998; Heskes, 2000; Martens & Grosse, 2015; Bernacchia et al., 2018), Generalized-Gauss-Newton matrix (Martens, 2014; Botev et al., 2017; Roux et al., 2008)[1]), which correspond to

$$\boldsymbol{H}_+\dot{\boldsymbol{w}} = \begin{bmatrix} w_2^\mathsf{T}\dot{\Delta} \\ \dot{\Delta} w_1^\mathsf{T} \end{bmatrix}. \tag{3}$$

This operation is indeed PSD, since $\dot{\boldsymbol{w}} \cdot \boldsymbol{H}_+\dot{\boldsymbol{w}} = \mathrm{Tr}[\dot{w}_1^\mathsf{T} w_2^\mathsf{T} \dot{\Delta} + \dot{w}_2^\mathsf{T} \dot{\Delta} w_1^\mathsf{T}] = \mathrm{Tr}[\dot{\Delta}\dot{\Delta}^\mathsf{T}] \geq 0$, where the dot-product denotes $\boldsymbol{a} \cdot \boldsymbol{b} \equiv \sum_{i=1}^d \mathrm{Tr}[a_i b_i^\mathsf{T}]$. We refer to this operation as Hessian$_+$.

**Null-space and Conservation laws** Deep linear networks exhibit inherent symmetries that their input-output map $\bar{w}$ is invariant under transformations that multiply arbitrary matrix $m$ to one layer and its inverse to the next layer, *i.e.* $\begin{bmatrix} w_1 \\ w_2 \end{bmatrix} \rightarrow \begin{bmatrix} mw_1 \\ w_2 m^{-1} \end{bmatrix}$, $\forall m$. $\dot{\boldsymbol{w}}_{\text{null}} \equiv \begin{bmatrix} mw_1 \\ -w_2 m \end{bmatrix}$ are the equivalent continuous-time transformations that yield the invariance $\dot{\Delta} = \dot{\bar{w}} = w_2 m w_1 - w_2 m w_1 = 0$, $\forall m$.

These transformations form the *null-space* of $\boldsymbol{H}_+$, since $\dot{\boldsymbol{w}}_{\text{null}} \cdot \boldsymbol{H}_+\dot{\boldsymbol{w}}_{\text{null}} = \mathrm{Tr}[\dot{\Delta}\dot{\Delta}^\mathsf{T}] = 0$, which is orthogonal to gradient, since $\dot{\boldsymbol{w}}_{\text{null}} \cdot \boldsymbol{g} = \mathrm{Tr}[\Delta \dot{\Delta}^\mathsf{T}] = 0$. Also orthogonal to the null-space is natural gradient, since $\dot{\boldsymbol{w}}_{\text{null}} \cdot \boldsymbol{H}_+^\dagger \boldsymbol{g} = \boldsymbol{g} \cdot \boldsymbol{H}_+^\dagger \dot{\boldsymbol{w}}_{\text{null}} = 0$, where $\boldsymbol{H}_+^\dagger$ denotes Moore-Penrose pseudo-inverse.

These continuous symmetries imply the following, self-explanatory theorem (Noether's theorem):

**Theorem 1** *All update rules $\dot{\boldsymbol{w}}$ that are orthogonal to the null-space, i.e.*

$$\dot{\boldsymbol{w}} \cdot \dot{\boldsymbol{w}}_{null} = \sum_{i=1}^d \mathrm{Tr}[(w_i \dot{w}_i^\mathsf{T} - \dot{w}_{i+1}^\mathsf{T} w_{i+1})m_i] = 0, \; \forall m_i$$

*exhibit the following conservation law*

$$d/dt\,(w_i w_i^\mathsf{T} - w_{i+1}^\mathsf{T} w_{i+1}) = 0, \; \forall i \tag{4}$$

This result was previously only known for gradient descent dynamics (Arora et al., 2018b; Du et al., 2018), which is generalized here.

# 3 LEARNING DYNAMICS

In this section, we analyze the learning dynamics of the network parameters $\boldsymbol{w}$ (Section 3.1) and the update dynamics of the input-output map $\bar{w}$ (Section 3.2) under a spectrum of curvature-corrected update rules. We then analyze how block-diagonal approximation modifies the curvature-corrected dynamics (Section 3.3).

## 3.1 PARAMETER DYNAMICS

We follow the singular value decomposition (SVD)-based analysis of Saxe et al. (2013); Advani & Saxe (2017); Lampinen & Ganguli (2018), by considering network weights that are initialized to

---

[1]Fisher matrix and Generalized-Gauss-Newton matrix are equivalent in many cases, including the least squares problem considered here (Pascanu & Bengio, 2013; Martens, 2014).

have their map's singular vectors aligned with those of $\bar{w}_*$.[2]Under such initialization, the update dynamics of weight matrices simplifies to their singular value dynamics, with their singular vectors remain unchanging. This simplified case admits exact analytic solutions, which provide good approximation to general learning dynamics. Moreover, this *aligned singular vector* condition is automatically satisfied for networks initialized with small random weights Saxe et al. (2013); Advani & Saxe (2017).

**Steepest gradient descent (SGD)**    Under SGD update, deep networks' weight parameters exhibit coupled nonlinear dynamics: ($d = 2$ example, $\eta$: learning rate)

$$\dot{\boldsymbol{w}} + \eta \boldsymbol{g} = \begin{bmatrix} \dot{w}_1 + \eta\, w_2^{\mathsf{T}}\, \Delta \\ \dot{w}_2 + \eta\, \Delta\, w_1^{\mathsf{T}} \end{bmatrix} = \mathbf{0}. \tag{5}$$

The SVD analysis decomposes eq (5) to individual singular mode dynamics. The dynamics of one singular mode is described by[3](See S.I.)

$$\dot{\sigma}_i + \eta\, \sigma_\Delta\, j_i = 0 \qquad\qquad (\sigma_\Delta = \bar{\sigma} - \bar{\sigma}_*, \; \bar{\sigma} = \prod_{i=1}^{d} \sigma_i) \tag{6}$$

where $\sigma_i, \bar{\sigma}_*, \bar{\sigma}, \sigma_\Delta$ are the singular values of $w_i, \bar{w}_*, \bar{w}, \Delta$, and $j_i \equiv \partial\bar{\sigma}/\partial\sigma_i = \bar{\sigma}/\sigma_i$ denotes the coupling between the input-output map and parameters, *i.e.* Jacobian. Note that this singular mode dynamics follows the hyperbolic paths

$$\sigma_i^2 - \sigma_k^2 = \text{constant}, \quad \forall i, k \tag{7}$$

which is the direct consequence of the conservation law (4). The *update speed* $\|\dot{\sigma}\|$ is proportional to the displacement $|\sigma_\Delta|$ and the coupling strength $\|j\|$

$$\|\dot{\sigma}\| \propto |\sigma_\Delta|\|j\|, \qquad (\|\dot{\sigma}\|^2 \equiv \sum_{i=1}^{d} \dot{\sigma}_i^2, \|j\|^2 \equiv \sum_{i=1}^{d} j_i^2) \tag{8}$$

which vanishes for networks with small coupling strength and explodes for large coupling strength.

**Natural gradient descent (NGD)**    NGD finds the minimum-norm update solution ($\min \dot{\boldsymbol{w}} \cdot \dot{\boldsymbol{w}}$) subject to the constraint (*i.e.* Moore-Penrose pseudo-inverse solution)

$$\boldsymbol{H}_+\dot{\boldsymbol{w}} + \eta \boldsymbol{g} = \begin{bmatrix} w_2^{\mathsf{T}}(\dot{\Delta} + \eta\Delta) \\ (\dot{\Delta} + \eta\Delta)w_1^{\mathsf{T}} \end{bmatrix} = \mathbf{0}, \tag{9}$$

which can be solved using Lagrange multipliers to yield (See S.I.)

$$\begin{bmatrix} \dot{w}_1 + \eta\, w_2^{\mathsf{T}}\, \Lambda \\ \dot{w}_2 + \eta\, \Lambda\, w_1^{\mathsf{T}} \end{bmatrix} = \mathbf{0}, \tag{10}$$

where $\Lambda$ satisfies

$$w_2^{\mathsf{T}} S(\Lambda) = S(\Lambda)w_1^{\mathsf{T}} = 0. \qquad (S(\Lambda) \equiv (w_2 w_2^{\mathsf{T}})\Lambda + \Lambda(w_1^{\mathsf{T}} w_1) - \Delta) \tag{11}$$

Remarkably, the only change from SGD update (5) is replacing $\Delta$ with $\Lambda$ as the main drive of dynamics eq (10), which preserves orthogonality to null-space and hence the conservation law (4) [4]. The singular mode dynamics of NGD update eq (10) is[5]

$$\dot{\sigma}_i + \eta\, \sigma_\Delta\, \frac{j_i}{\|j\|^2} = 0, \tag{12}$$

where $\sigma_\Delta$ of SGD eq (6) is replaced by $\sigma_\Lambda = \sigma_\Delta/\|j\|^2$, the singular values of $\Lambda$ (See S.I.). NGD dynamics eq (12) follows the same hyperbolic paths of SGD eq (7), but with modified *update speed*

$$\|\dot{\sigma}\| \propto \frac{|\sigma_\Delta|}{\|j\|} \tag{13}$$

which inversely scales with $\|j\|$. Therefore, NGD's update speed explodes for small coupling strength, reciprocal to SGD's vanishing speed phenomenon.

---

[2]Given SVD of weight matrices $w_i = L_i D_i R_i^{\mathsf{T}}$ and $\bar{w}_* = L_* D_* R_*^{\mathsf{T}}$, where $D$ are the diagonal singular value matrices and $L/R$ are the left/right singular vector matrices, the *aligned singular vector condition* assumes $R_1 = R_*, L_d = L_*$ and $R_{i+1} = L_i$ for all layers $1 \le i \le d - 1$.

[3] The dynamics eq (6),(12) apply to all *active* singular modes. *Inactive* modes that have $\bar{\sigma} = 0$ stay frozen. The number of active modes is determined by the bottleneck size, *i.e.* the narrowest width of network.

[4]The Moore-Penrose pseudo-inverse solution is guaranteed to be orthogonal to the null-space, since non-zero null-space components only increase the solution's norm without affecting the constraint eq (9).

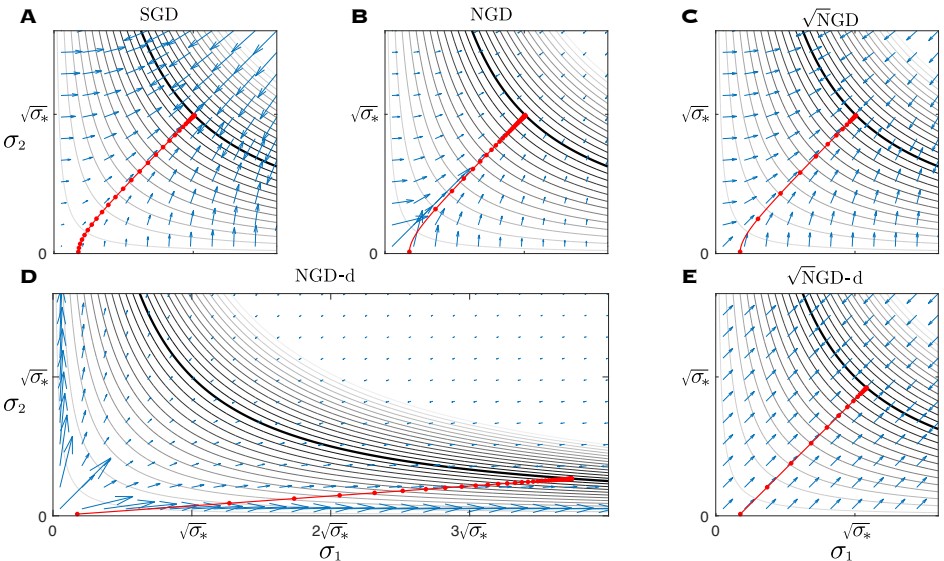

Figure 1: Learning dynamics of a singular mode of a single hidden layer network ($d = 2$). The contour lines visualize the manifolds of constant displacement levels $\sigma_\Delta \equiv \sigma_1 \sigma_2 - \bar{\sigma}_*$. The optimal solution $\sigma_\Delta = 0$ is shown in black. Tangent space of the manifolds defines the null-space of Hessian$_+$. The vector field visualizes the *displacement-normalized update* $[\dot{\sigma}_1, \dot{\sigma}_2]/|\sigma_\Delta|$, *whose amplitude is the normalized update speed*: $\|\dot{\sigma}\|/|\sigma_\Delta| \propto \|j\|^{1-2/q}$. (A,B,C) SGD, NGD, $\sqrt{}$NGD share the same update directions defined by the hyperbolic paths that conserve $\sigma_1^2 - \sigma_2^2$ (red lines), orthogonal to the null-space. But they exhibit different *update speed*: SGD exhibits vanishing speed problem for small weights, while NGD has the opposite problem. In contrast, $\sqrt{}$NGD exhibits constant *normalized speed*. (D) NGD-d exhibits radially diverging vector field that conserves $\sigma_1/\sigma_2$. The learning trajectories of NGD and NGD-d traverse the contour lines with synchronized timing (red dots), (E) $\sqrt{}$NGD-d exhibits vector field of constant direction and amplitude that conserves $|\sigma_1| - |\sigma_2|$.

**Fractional Natural Gradient Descent ($\sqrt[q]{\textbf{NGD}}$)**   Above results can be generalized to a spectrum of update rules that apply *partial* curvature corrections, described by $\sqrt[q]{H_+}\dot{w} + \eta g = 0$, where $\sqrt[q]{H_+}$ is a fractional power of Hessian$_+$ ($q \geq 1$). The singular mode dynamics of $\sqrt[q]{}$NGD is

$$\dot{\sigma}_i + \eta\, \sigma_\Delta\, \frac{j_i}{\|j\|^{2/q}} = 0, \tag{14}$$

which interpolates between NGD ($q = 1$) and SGD ($q \to \infty$). Eq (14) follows the same hyperbolic paths of SGD eq (7), but with modified update speed

$$\|\dot{\sigma}\| \propto |\sigma_\Delta| \|j\|^{1-2/q}. \tag{15}$$

Note that for $q = 2$, termed $\sqrt{}$NGD, the update speed becomes independent of the coupling strength

$$\|\dot{\sigma}\| = \eta\, |\sigma_\Delta|, \tag{16}$$

thus eliminating the vanishing/exploding update speed problems of SGD/NGD (See Fig 1C).

**Relation to Regularized NGD**   Alternative interpolation solves $(H_+\dot{w} + \eta g) + \epsilon I(\dot{w} + \eta g) = 0$ ($\epsilon \geq 0$), which yields the regularized (or damped) inverse

$$\dot{w} = -\eta(\epsilon + 1)(\epsilon I + H_+)^{-1}g, {}^{6} \tag{17}$$

similar to Levenberg-Marquardt damping (less the $(\epsilon + 1)$ term), whose singular mode dynamics is

$$\dot{\sigma}_i + \eta\, \sigma_\Delta\, \frac{j_i}{\|j\|}\left(\frac{a\|j\| + 1}{a + \|j\|}\right) = 0, \qquad (a \equiv \epsilon/\|j\|) \tag{18}$$

---

[6] This expression reduces to SGD in the limit $\epsilon \to \infty$, which differs from the usual regularized inverse $\dot{w} = -\eta(\epsilon I + H_+)^{-1}g$, which reduces to $\mathbf{0}$.

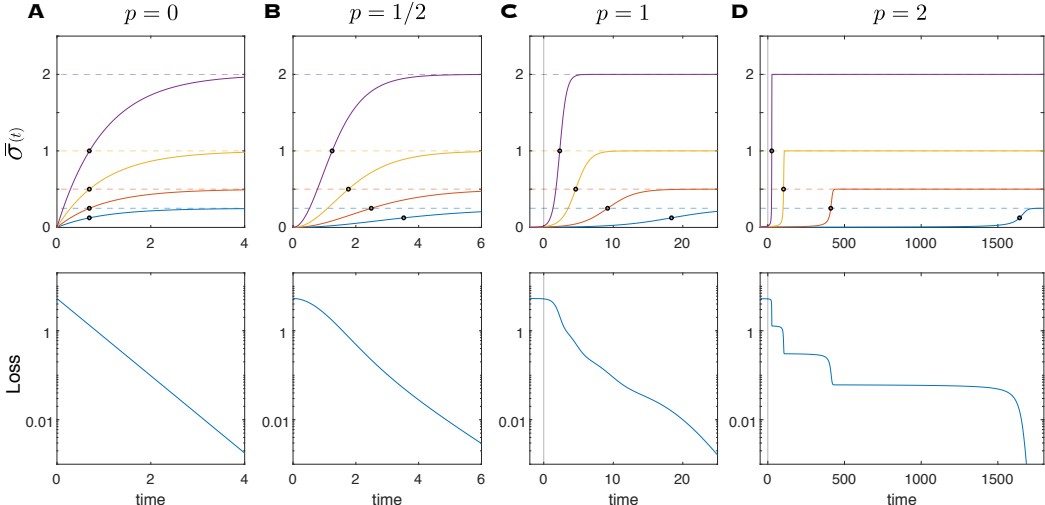

Figure 2: Learning curves of input-output map and loss profile for various stiffness $p$ levels: $p = 0$ corresponds to NGD update, $1/2 \leq p < 1$ corresponds to $\sqrt{\text{NGD}}$ update, and $1 \leq p < 2$ corresponds to SGD update for network depth ranging between $2 \leq d < \infty$. Top: Learning curves of map singular modes $\bar{\sigma}_{(t)}$ from eq (21). Dashed lines show the mode-strength of dataset $\bar{\sigma}_*$. Note that large $p$ increases the stiffness of dynamics, *i.e.* extreme changes of time-scale: between extreme slow and extreme fast. Half-max points (black circles) are shown to visualize the overall time-scale of learning dynamics, which decreases with mode strength as $\bar{\sigma}_*^{-p}$. Bottom: Corresponding loss profiles. Initial conditions: $\bar{\sigma}_{(0)} = 0$ for $p < 1$, and $\bar{\sigma}_{(0)} = \bar{\sigma}_*/100$ for $p \geq 1$. $\bar{\eta} = 1$.

where the ratio $a \equiv \epsilon/\|j\|$ describes the effective degree of interpolation between NGD ($a \to 0$) and SGD ($a \to \infty$). Note that $a$ should be large enough to provide sufficient damping, but not too large to nullify the effect of curvature correction, which is difficult to simultaneously satisfy across all singular modes with fkxed $\epsilon$. $\sqrt{\text{NGD}}$ can be considered as providing ideally and adaptively tuned regularization ($a = 1$) for all singular modes, where the regularization is most effective.

## 3.2 MAP DYNAMICS

The parameter update ultimately drives the learning dynamics of the input-output map, via Jacobian

$$\dot{\bar{\sigma}} = \sum_{i=1}^{d} \dot{\sigma}_i \, j_i, \tag{19}$$

which yields the following map learning dynamics under $\sqrt[q]{\text{NGD}}$ update (14)

$$\dot{\bar{\sigma}} = -\eta(\bar{\sigma} - \bar{\sigma}_*)\|j\|^{2(1-1/q)}. \tag{20}$$

In general, eq (20) does not admit closed-form solutions due to the coupling strength term, with the exception of NGD ($q = 1$). As shown by the vector field in Figure 1, however, the coupling strength changes in a streotypical manner along the learning trajectories. Therefore, the general characteristics of map dynamics can be appreciated from the representative case of balanced weights: $\sigma_i = \bar{\sigma}^{1/d} \ \forall i$, or in terms of the conserved quantities, $w_i w_i^\mathsf{T} - w_{i+1}^\mathsf{T} w_{i+1} = 0$. Note that this balanced weight condition is automatically approximately satisfied if the networks are initialized with small random weights.

Under the balanced weight condition, eq (20) simplifies to

$$\dot{\bar{\sigma}} = -\bar{\eta} \, (\bar{\sigma} - \bar{\sigma}_*) \, \bar{\sigma}^p \qquad \left( p \equiv \frac{2(d-1)(q-1)}{d \, q} \right) \tag{21}$$

where $\bar{\eta} \equiv \eta \, d^{1-1/q}$ is the *depth-calibrated* learning rate, and $p$ represents the combined effect of depth and curvature correction that determines the *stiffness*, or degree of nonlinearity, of map

dynamics. Figure 2 shows the following notable closed-form solutions, as well as $p = 2$ case:

$$\bar{\sigma}_{(t)} = \bar{\sigma}_*(1 - e^{-\bar{\eta}t}) \qquad\qquad (\,p = 0\,)$$

$$\bar{\sigma}_{(t)} = \bar{\sigma}_* \tanh^2(\bar{\eta}\sqrt{\bar{\sigma}_*}t/2) \qquad\qquad (\,p = 0.5\,)$$

$$\bar{\sigma}_{(t)} = \frac{\bar{\sigma}_*}{1 + (\bar{\sigma}_*/\bar{\sigma}_{(0)} - 1)e^{-\bar{\eta}\bar{\sigma}_*t}} \qquad\qquad (\,p = 1\,)$$

where zero initial condition $\bar{\sigma}_{(0)} = 0$ is assumed for $p < 1$ cases.

**NGD update ($q = 1, p = 0$)** Under NGD update, the map dynamics exhibits fully linearized convergence dynamics with a constant time-scale $\eta^{-1}$ for all depth $d$ and data mode-strength $\bar{\sigma}_*$. Its learning curves exhibit finite growth rate near zero $\bar{\sigma}_{(t)} \approx \bar{\eta}\,\bar{\sigma}_* t$, which entails exploding parameter update speed as the coupling strength approaches zero. Therefore, the full curvature correction of NGD sacrifices stability of parameter dynamics in order to perfectly cancel out all nonlinearities of map dynamics.

**$\sqrt{\textbf{NGD}}$ update ($q = 2$, $p = 1 - 1/d$)** For $\sqrt{\text{NGD}}$ update, the stiffness ranges from $p = 0.5$ for single hidden layer networks to $p \to 1$ in infinite depth limit. Its learning curves exhibit polynomial growth near zero, $\bar{\sigma}(t) \propto t^{1/(1-p)}$, which takes finite time to escape from zero initial condition. even though the initial growth rate vanishes with the coupling strength. The overall time-scale of learning decreases with mode strength as $\bar{\sigma}_*^{-p}$, such that stronger singular modes (large $\bar{\sigma}_*$) learn faster than weaker modes.

**SGD update ($q \to \infty$, $p = 2 - 2/d$)** Under SGD update, the stiffness ranges from $p = 1$ for single hidden layer networks to $p \to 2$ in infinite depth limit. Its learning curves exhibit sigmoidal shape that take infinite time to escape from the saddle point at zero initial condition: the escape time diverges as $\mathcal{O}(-\log \bar{\sigma}_{(0)})$ for $p = 1$ and $\mathcal{O}(\bar{\sigma}_{(0)}^{1-p})$ for $p > 1$. Also, the increased $p$ causes greater separation of time-scales $(\bar{\eta}\bar{\sigma}_*^p)^{-1}$ across singular modes, which results in stage-like transitions over the course of training, with each singular mode making sudden transition from slow learning to rapid convergence (Saxe et al., 2013).

**Effective Depth** Network depth $d$ and curvature correction $q$ interact in a symmetric manner, which can be intuitively understood by representing stiffness in terms of the corresponding network depth under SGD update, called the *effective depth*:

$$d_{\text{eff}} = \frac{dq}{d + q - 1}, \qquad\qquad (22)$$

which approaches the actual depth $d_{\text{eff}} \to d$ in the SGD limit ($q \to \infty$), and similarly, approaches $d_{\text{eff}} \to q$ in the limit of infinite depth ($d \to \infty$). Therefore, $\sqrt[q]{\text{NGD}}$ reduces the network's effective depth to be strictly less than $q$. For $\sqrt{\text{NGD}}$, this upper-limit is 2, *i.e.* single hidden layer network.

To summarize, curvature correction lowers the nonlinearity/stiffness of map dynamics of deep networks by reducing their effective depth. The full curvature correction effect of NGD perfectly cancels out all nonlinearities of map dynamics to exhibit linear convergence, equivalent to shallow network learning, but it sacrifices stability of parameter dynamics to explode at the saddle point. In contrast, partial curvature correction of $\sqrt{\text{NGD}}$ directly facilitates the parameter update dynamics, which eliminates the vanishing/exploding update problem, and it makes the map dynamics only slightly nonlinear, but no more so than that of single hidden layer networks under gradient descent.

## 3.3 EFFECT OF LAYER-RESTRICTED APPROXIMATION

**Block-diagonal NGD (NGD-d)** In most practical deep learning applications, numerically estimating and inverting Hessian$_+$ becomes prohibitively expensive. Instead, most second-order methods approximate NGD by applying *layer-restricted* curvature corrections, ignoring the off-block-diagonal Hessian$_+$ terms across different layers (Martens & Grosse, 2015; Ba et al., 2016; Grosse & Martens, 2016; Martens et al., 2018; Bernacchia et al., 2018): ($d = 2$ example)

$$\begin{bmatrix} H_1\dot{w}_1 + \eta_1 g_1 \\ H_2\dot{w}_2 + \eta_2 g_2 \end{bmatrix} = \begin{bmatrix} w_2^{\intercal}w_2\dot{w}_1 + \eta_1 w_2^{\intercal}\Delta \\ \dot{w}_2 w_1 w_1^{\intercal} + \eta_2 \Delta w_1^{\intercal} \end{bmatrix} = \mathbf{0}, \qquad\qquad (23)$$

which nevertheless satisfies the NGD constraint (9) if $\sum_{i=1}^{d} \eta_i = \eta$. $H_i$ denotes the block-diagonal Hessian$_+$ term of layer $i$. Singular mode dynamics of eq (23) is (with $\eta_i = \eta/d$)

$$\dot{\sigma}_i + \frac{\eta\,\sigma_\Delta}{d\,j_i} = 0, \tag{24}$$

where the layer-restricted factor $j_i^2$ substitutes the full curvature correction factor $\|j\|^2$ of NGD (12).

This block-diagonalization significantly modifies the parameter update dynamics by adding non-zero null space components. Instead of the hyperbolic paths (7), eq (24) follows radially diverging paths that conserve $\sigma_i/\sigma_k$ as constants of motion. Consequently, NGD-d update exhibits larger parameter update speed than NGD[7], and converges to less efficient, large norm solutions that are highly sensitive to initial conditions and perturbations (Fig 1D, red line). Despite the vastly different parameter dynamics, however, NGD-d exhibits identical map learning dynamics as NGD $\dot{\bar{\sigma}} = -\eta\,(\bar{\sigma} - \bar{\sigma}_*)$ (Fig 1BD, red dots), because the input-output map is invariant under null-space transformations.

**Block-diagonal $\sqrt{\text{NGD}}$ ($\sqrt{\text{NGD}}$-d)** More generally, block-diagonalized fractional NGD $H_i^{1/q}\dot{w}_i + \eta\,g_i/d^{1/q} = 0$ yields

$$\dot{\sigma}_i + \frac{\eta\,\sigma_\Delta}{d^{1/q}}\,j_i^{1-2/q} = 0, \tag{25}$$

which conserves $\sigma_i^{2(1-1/q)} - \sigma_k^{2(1-1/q)}$ as constants of motion. For $q = 2$, called $\sqrt{\text{NGD}}$-d, the singular mode dynamics

$$\dot{\sigma}_i + \frac{\eta\,\sigma_\Delta}{\sqrt{d}}\,\text{sign}(j_i) = 0, \tag{26}$$

follows non-diverging, parallel paths that conserve $|\sigma_i| - |\sigma_k|$, with identical parameter update speed as $\sqrt{\text{NGD}}$'s $\|\dot{\sigma}\| = \eta\,|\sigma_\Delta|$ (Fig 1E). Therefore, $\sqrt{\text{NGD}}$-d yields neutrally-stable update dynamics that neutralizes the vanishing/exploding update speed problems.

## 4 IMPLICIT BIAS FOR REGULARIZATION

Recent works have shown that learning dynamics of deep networks under SGD update exhibits implicit bias towards keeping the network well regularized. Here, we consider two such properties, and analyze how they generalize under curvature-corrected update rules.

**Weight balance** Deep neural networks often exhibit redundant parameterizations, such that configurations of parameters implement the same input-output map. One such redundancy, or symmetry, that concerns both deep linear networks and deep ReLU networks is homogeneity, such that multiplying a layer by a positive scalar $c$ and divide another layer by $c$, does not change the input-output map. The problem is that $c$ can be arbitrarily large or small, yielding potentially unbounded, yet valid solutions. Such unboundedness poses major theoretical difficulty for convergence analysis of gradient-based local optimization methods (Lee et al., 2016; Shamir, 2018).

Fortunately, SGD update exhibits implicit bias toward automatically balancing the norm of different layer's weight matrices. The proof directly follows from the following conserved quantity of scalar multiplication symmetry $\|w_i\|_{\text{Frob}}^2 - \|w_{i+1}\|_{\text{Frob}}^2$, which is a relaxed version of the aforementioned conservation law eq (4). Thus, if the weights are initially small, this differences between squared norm will remain small through out learning process, thus establishing balancedness across layers Du et al. (2018).

As shown in section 2, curvature-corrected updates (*e.g.* NGD and $\sqrt{\text{NGD}}$) retain orthogonality to the null-space of symmetry, and thus comply with the same conservation laws as SGD. We show numerical confirmation of this prediction in S.I. The conservation of squared difference of norms for homogeneous ReLU networks still requires similar numerically confirmation.

In contrast, block-diagonalized methods do not follow the same conservation law. NGD-d conserves the ratio between singular values across layers $\sigma_i/\sigma_k$, which does not guarantee balancedness even

---

[7] $\|\dot{\sigma}\|_{\text{NGD-d}}^2 \geq \|\dot{\sigma}\|_{\text{NGD}}^2$ can be shown using Jensen's inequality: $\frac{1}{d}\sum_{i=1}^{d}\frac{1}{j_i^2} \geq \frac{d}{\sum_{i=1}^{d} j_i^2}$,

with small initialization. $\sqrt{\text{NGD}}$-d, however, conserves the absolute difference of singular values across layers $|\sigma_i| - |\sigma_k|$, which guarantees balancedness, at least under the condition of aligned singular vectors: That is, the ratio between the singular values would approach close to 1 if they grow from small initial values, while maintaining the small absolute difference. Although this does not constitute a formal proof for general case, $\sqrt{\text{NGD}}$-d confirms to maintain balancedness across layers in numerical simulations (See S.I.).

**Low rank approximation / Generalization dynamics**   The learning dynamics of the input-output map under SGD update separates the time-scales of learning across singular modes $(\bar{\eta}\bar{\sigma}_*^P)^{-1}$, such that the singular modes with stronger data correlation preferentially are learned faster (Saxe et al., 2013). This property yields an implicit regularization property for deep networks to efficiently extract low-rank structure of the dataset, such as for finding matrix factorizations with minimum nuclear norm (Gunasekar et al., 2017; Arora et al., 2019). It also allows deep networks to avoid overfitting via early stopping by first learning the signal dimensions of noisy dataset, before the overfitting of the noise dimensions occurs, as long as signal-to-noise ratio is sufficiently large Advani & Saxe (2017); Lampinen & Ganguli (2018) which yields good generalization performance to unseen data. However, this approach requires the network to be trained from small random weight initialization, where SGD suffers from vanishing gradient problem.

In curvature-corrected cases, the learning speed of map dynamics eq (21) scales as $\bar{\sigma}_*^{-p}$. Under NGD, the map dynamics is perfectly linearized ($p = 0$), which also removes its ability to separate out the time-scales. This makes NGD update prone to large generalization error due learning the noise dimension simultaneously with the signal. In contrast, $\sqrt{\text{NGD}}$ partially retains time-scale separation in the learning dynamics, while also accelerating the parameter update dynamics near zero weights.

We the generalization property of curvature corrected learning rules with student-teacher task from Lampinen & Ganguli (2018), in which the training and test dataset are generated by a teacher network $y^\mu = \bar{w}_* x^\mu + z^\mu$, where $x^\mu \in \mathbb{R}^N$ is the input data, $y^\mu \in \mathbb{R}^N$ is the output, $\bar{w}_* x^\mu$ is the signal and $z^\mu \in \mathbb{R}^N$ is the noise. Teacher's input-output map $\bar{w}_* \in \mathbb{R}^{N \times N}$ is assumed to have a low-rank structure (rank 3), and the student is a depth $d = 4$ network of constant width $N = 16$, whose weight matrices are initialized to have maximum singular value of $0.05$. The number of training dataset $\{x^\mu, y^\mu\}_{\mu=1}^P$ is set to be equal to the effective number of parameters $P = N$, which makes the learning process most susceptible to overfitting.

For numerical calculation of NGD and $\sqrt{\text{NGD}}$, the Hessian$_+$ block between layer $i$ and $k$ are computed as described in Bernacchia et al. (2018) (eq 42), which are then concatenated to the full Hessian$_+$ matrix and numerically inverted (or sqrt-inverted) via eigen-decomposition. Levenberg-Marquardt damping of $\epsilon = 10^{-5}$ and update clipping are used for numerical stability of NGD. $\sqrt{\text{NGD}}$ does not require such clipping or damping terms.

Figure 3 shows the result of training. SGD exhibits stage-like transitions, which first learns the three signal modes, well separated from the onset of overfitting of the noise modes begins, which allows effective early stopping scheme. However, it suffers from the long plateaus due to vanishing gradient problem.

NGD (and NGD-d) updates learn all singular modes simultaneously including the noise modes (See Fig 3 D), which leads to high generalization error. Note that NGD's loss profile deviates from exponential decay due to the clipping. In contrast, $\sqrt{\text{NGD}}$ (and $\sqrt{\text{NGD}}$-d) allows fast learning while separating the signal dimensions from the noise dimensions, achieving comparable test loss as SGD update, but also with fast early-stopping time comparable to NGD update. Note that all three update rules achieve the same test loss after overfitting is complete. This is due to the shared learning path for each singular mode across the methods.

## 5   CONCLUSION

To summarize our contribution, we derived a generalized conservation law that describes the optimization paths of network parameters under gradient descent as well as curvature-corrected update rules. Consequently, curvature correction only affects the speed of convergence without affecting other qualitative properties of parameter update process.

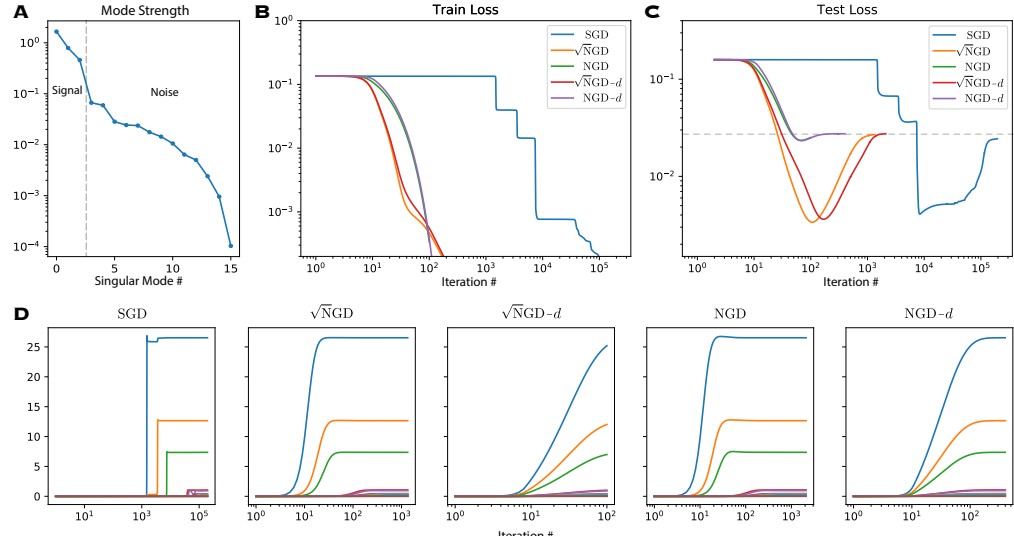

Figure 3: Curvature correction effect on generalization dynamics: (A) Singular mode strength of input-output correlation of training dataset. Dataset is generated from a rank-3 teacher network with added noise (SNR = 10). (B, C) Training and testing loss profiles of a 3-hidden-layer student network (See text). Note that all methods eventually converge to the identically overfitted solution (horizontal dashed-line). (D) Time-separated learning dynamics of across singular modes. To obtain the individual mode components, we computed the network's input-output correlation matrix, and projected it to the singular vector basis of the data correlation matrix.

We revealed a trade-off between map dynamics and parameter dynamics: The full curvature correction effect of natural gradient descent (NGD) completely linearizes the map learning dynamics of deep networks, equivalent to that of shallow networks. Such complete linearization, however, sacrifices stability of parameter update dynamics to explode when gradient vanishes and vice versa. Moreover, we introduced $\sqrt{\text{NGD}}$ that partially corrects for the effect of curvature, which facilitates the parameter update dynamics by eliminating the vanishing/exploding update problems. This makes the map dynamics slightly nonlinear, but no more so than that of single hidden layer networks under gradient descent. Moreover, NGD makes the learning process prone to overfitting by simultaneously learning both the signal and the noise dimensions of data, whereas $\sqrt{\text{NGD}}$ partially retains gradient descent's resistance to overfitting by separating the time-scales between the signal and the noise dimensions. We also showed that the widely-used block-diagonal approximation of NGD breaches the aforementioned conservation law, resulting in highly divergent parameter update dynamics, which breaks the weight balance across layers. In contrast, block-diagonalization of $\sqrt{\text{NGD}}$ preserves stability of parameter update dynamics, yielding efficient and stable learning algorithms.

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

# Supplemental Materials

## S.I.1    MOORE-PENROSE INVERSE SOLUTION: EQ (15,16,17) IN SECTION 3.2

In section 4, we find the Moore-Penrose inverse solution that minimizes the update norm $\dot{\boldsymbol{w}} \cdot \dot{\boldsymbol{w}} = \sum_i \text{Tr}[\dot{w}_i \dot{w}_i^\mathsf{T}]$ while satisfying the natural gradient constraint: ($d = 2$ example)

$$\boldsymbol{H}\dot{\boldsymbol{w}} + \eta\boldsymbol{g} = \begin{bmatrix} w_2^\mathsf{T}(\dot{\Delta} + \eta\Delta) \\ (\dot{\Delta} + \eta\Delta)w_1^\mathsf{T} \end{bmatrix} = \boldsymbol{0}. \tag{S.I.1}$$

This constrained optimization problem is described by the following Lagrangian:

$$\mathcal{L}(\dot{w}_1, \dot{w}_2, \Lambda_2, \Lambda_2) = (\dot{w}_1 \cdot \dot{w}_1 + \dot{w}_2 \cdot \dot{w}_2)/2 + \Lambda_1 \cdot w_2^\mathsf{T}(\dot{\Delta} + \eta\Delta) + \Lambda_2 \cdot (\dot{\Delta} + \eta\Delta)w_1^\mathsf{T},$$

where $\dot{\Delta} = w_2\dot{w}_1 + \dot{w}_2 w_1$, and dot notation denotes inner-product: $a \cdot b \equiv \text{Tr}[a^\mathsf{T} b]$. Optimality condition on $\dot{w}_i$ yields

$$\partial\mathcal{L}/\partial\dot{w}_1 = \dot{w}_1 + w_2^\mathsf{T} w_2 \Lambda_1 + w_2^\mathsf{T}\Lambda_2 w_1 = 0 \tag{S.I.2}$$
$$\partial\mathcal{L}/\partial\dot{w}_2 = \dot{w}_2 + w_2\Lambda_1 w_1^\mathsf{T} + \Lambda_2 w_1 w_1^\mathsf{T} = 0 \tag{S.I.3}$$

which, via change of variables $\Lambda \equiv (w_2\Lambda_1 + \Lambda_2 w_1)/\eta$, reduces to

$$\dot{w}_1 + \eta w_2^\mathsf{T}\Lambda = 0 \tag{S.I.4}$$
$$\dot{w}_2 + \eta\Lambda w_1^\mathsf{T} = 0 \tag{S.I.5}$$

which can be plugged into the optimality condition on $\Lambda_i$

$$\partial\mathcal{L}/\partial\Lambda_1 = w_2^\mathsf{T}(\dot{\Delta} + \eta\Delta) = 0 \tag{S.I.6}$$
$$\partial\mathcal{L}/\partial\Lambda_2 = (\dot{\Delta} + \eta\Delta)w_1^\mathsf{T} = 0 \tag{S.I.7}$$

to produce a linear equation for $\Lambda_i$:

$$w_2^\mathsf{T} S(\Lambda) = S(\Lambda)w_1^\mathsf{T} = 0 \tag{S.I.8}$$
$$\text{where} \quad S(\Lambda) = (w_2 w_2^\mathsf{T})\Lambda + \Lambda(w_1^\mathsf{T} w_1) - \Delta. \tag{S.I.9}$$

Note that if $w_2$, $w_1$ are invertible, it is easy to see that eq (S.I.6)(S.I.7) reduce to exponentially converging dynamics $\dot{\Delta} + \eta\Delta = \dot{\bar{w}} + \eta(\bar{w} - \bar{w}^*) = 0$, with the solution of $S(\Lambda) = 0$ driving the parameter update eq (S.I.4)(S.I.5). This result also holds true for the over-complete cases, where the hidden layer width is larger than the minimum of input layer or output layer size. For the under-complete cases, *i.e.* with bottleneck hidden layers, the exponential convergence applies only to the subspace dimensions permitted by the bottleneck, with the other dimensions remain frozen.

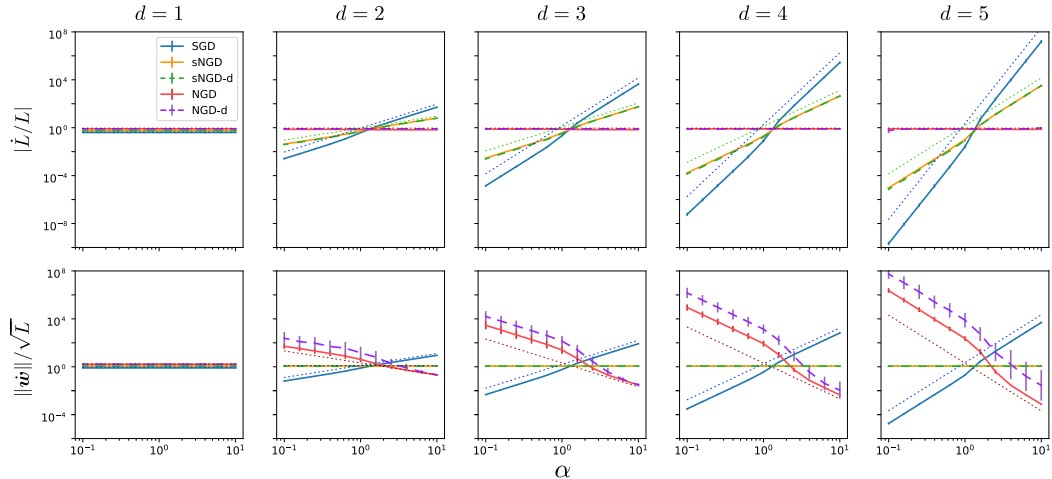

Figure S.I.1: Scaling relationships for normalized loss update $\dot{L}/L$ and weight/parameter update speed $\|\dot{\boldsymbol{w}}\|/\sqrt{L}$ shown across different depth $d$ and curvature correction factor $q$. $\alpha$ is the overall scale factor for the weights. Dotted-lines show predictions for the case of balanced weights, eq (S.I.16)(S.I.17). The numerical results show the result from networks of random weights sampled from Gaussian distribution, (Xavier normalization (Glorot & Bengio, 2010)), They exhibit the same power-law exponent as the dotted-lines (*i.e.* same slope), but shifted toward right. Block-diagonal approximation does not affect the power-law exponents, but it affects the coefficient such that NGD-d exhibits much larger parameter update size than NGD, while exhibiting identical zero power-law for loss update. In contrast, sNGD-d exhibits identical match with sNGD for both parameter update and loss update. See S.I. for detailed setup. Error bars show standard deviation from 10 simulations with random weights.

## S.I.2    SINGULAR MODE ANALYSIS EQ (18) IN SECTION 3.2

We follow the SVD-base analysis under the aligned singular vector condition (Saxe et al., 2013). We introduce $\sigma_i, \bar{\sigma}, \sigma_\Delta, \sigma_\Lambda, \sigma_S$ which represent the singular values of $w_i, \bar{w}, \Delta, \Lambda, S(\Lambda)$ of one particular singular mode, and $j_i \equiv \frac{\partial \bar{\sigma}}{\partial \sigma_i} = \prod_{k \neq i} \sigma_k$ In this representation, eq (S.I.4)(S.I.5) reduce to

$$\dot{\sigma}_i = -\sigma_\Lambda j_i \tag{S.I.10}$$

whereas, eq (S.I.8)(S.I.9) reduce to

$$\sigma_i \sigma_S = 0 \tag{S.I.11}$$

$$\sigma_S = \sum_{i=1}^{d} j_i^2 \sigma_\Lambda - \eta \sigma_\Delta \tag{S.I.12}$$

An *active singular* mode should have at least one non-zero $\sigma_i$, which according to eq (S.I.11), implies $\sigma_S = 0$. Therefore, from eq (S.I.12),

$$\sigma_\Lambda = \eta \frac{\sigma_\Delta}{\sum_{i=1}^{d} j_i^2} \tag{S.I.13}$$

which plugs into (S.I.10) to produce the result in the main text

$$\dot{\sigma}_i = -\eta \frac{\sigma_\Delta}{\sum_{i=1}^{D} j_i^2} \tag{S.I.14}$$

## S.I.3    SCALING LAWS OF UPDATE DYNAMICS

The parameter update (14) and map learning dynamics (20) of a singular mode exhibit the following scaling relationships with respect to the coupling strength $\|j\|$:

$$\|\dot{\sigma}\|/|\sigma_\Delta| \propto \|j\|^{1-2/q}, \qquad\qquad \dot{\bar{\sigma}}/\sigma_\Delta \propto -\|j\|^{2-2/q}, \tag{S.I.15}$$

defined for individual singular modes under the well-aligned singular vector condition.

For the *balanced* weight case where the coupling strength term is constant across all singular modes, eq (S.I.15) reduces to more generally applicable scaling relationships:

$$\|\dot{\boldsymbol{w}}\|/\sqrt{L} \propto \alpha^{(d-1)(1-2/q)} \tag{S.I.16}$$

$$|\dot{L}/L| \propto \alpha^{(d-1)(2-2/q)}, \tag{S.I.17}$$

where $\alpha$ is the overall scale factor for the weight matrices. Instead of the detailed description of individual singular mode dynamics, eq (S.I.16)(S.I.17) encapsulate the overall scaling law between weights and update speed that can be readily measured without requiring the aligned singular vector condition.

Under SGD ($q \to \infty$), weight update scales as $\alpha^{d-1}$, and double the power for the loss update, exhibiting the vanishing update problem for small $\alpha$. Under sNGD ($q = 1/2$), weight update is constant (zero power-law) with respect to $\alpha$ and loss update scales as $\alpha^{d-1}$. Under NGD ($q = 1$), loss update is constant (zero power-law) with respect to $\alpha$, but the weight update inversely scales as $\alpha^{-(d-1)}$, exhibiting the exploding update problem for small $\alpha$.

Numerical experiments indeed confirm these scaling laws (Figure S1): For the case of balanced weights the predictions hold exactly, and approximately for random weight matrices. Note that the zero-power law predictions for weight update under sNGD and for loss update under NGD are exact, because they indeed satisfy the *balanced* condition: the coupling strength term with zero exponent $\|j\|^0$ is indeed constant.

## 4 NUMERICAL CONFIRMATION OF CONSERVATION LAW

See Figure S2. Figure S2 plots the learning trajectory of a 3 layer network, and shows the elements of the weight matrices evolving over time ($w_1, w_2, w_3$). It also shows the conserved quantities $w_1 w_1^\mathsf{T} - w_2^\mathsf{T} w_2$, $w_2 w_2^\mathsf{T} - w_3^\mathsf{T} w_3$, which indeed remain constant for SGD, NGD and $\sqrt{\text{NGD}}$, while it blows up for NGD-d. $\sqrt{\text{NGD}}$ also violates the conservation law, but the weights remain balanced over time.

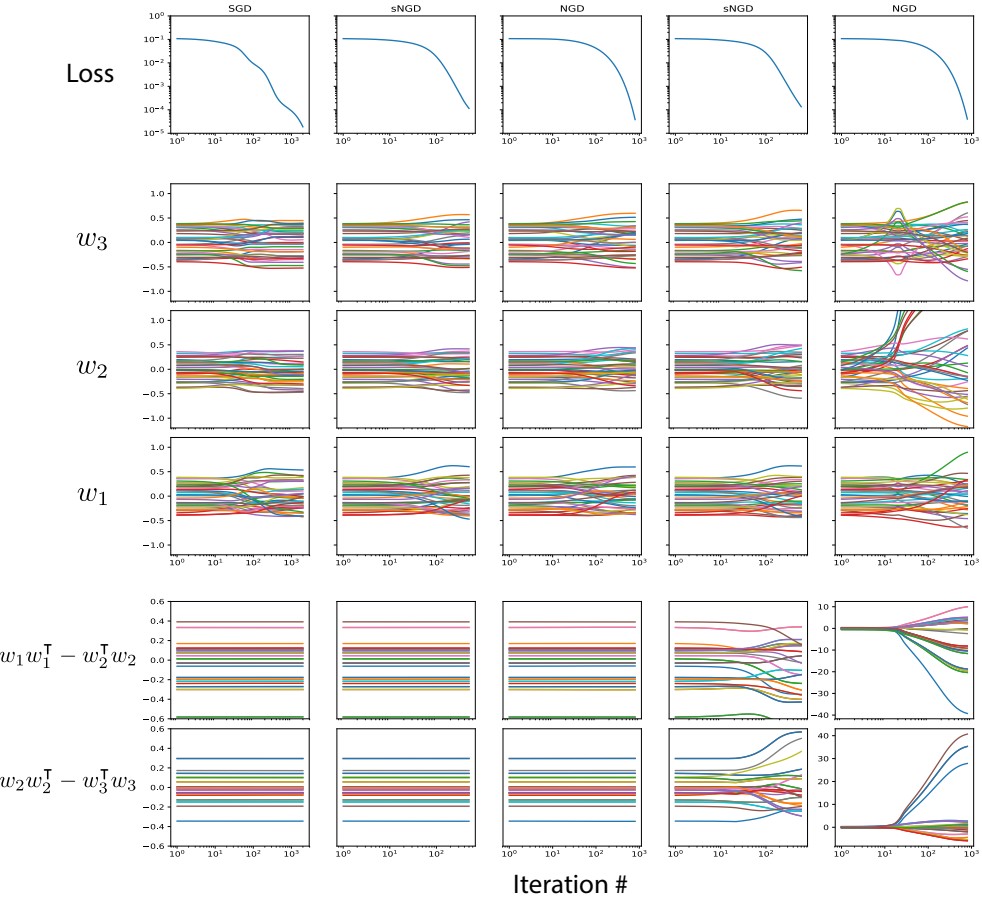

Figure 2: Training of a linear 3-layer network ($D = 2$).

