# OpenReview forum: "EXACT ANALYSIS OF CURVATURE CORRECTED LEARNING DYNAMICS IN DEEP LINEAR NETWORKS"
_ICLR.cc/2020/Conference — Reject_

### Official Review · AnonReviewer4 · 2019-10-23
**Official Blind Review #4**

**Rating:** 6

**Review:**

In this paper, the authors study the training dynamics of natural gradient descent with linear DNNs.
Specifically, they showed that the curvature corrected natural gradient descents preserve the learning trajectory of plain gradient descent and only affect the temporal dynamics. A fractional natural gradient
descent method that only applies partial curvature correction is proposed to address the numerical stability
issue of vanilla natural gradient descent.

The paper is well presented and the derivations of analytical solutions are clear. Although the analysis is
limited to the linear case, it does give some insight into the behaviors of curvature corrected gradient descents.
However, it would be interesting to see how these algorithms would perform under the no-linear case. Also,
will these results hold for other neural network structures? For example, the Hessian of each layer of
ResNet will be close to orthogonal and how will that affect the NGD?

Overall I think this is an interesting paper with strong results and vote for accepting.

**Experience Assessment:**

I do not know much about this area.

**Review Assessment: Checking Correctness Of Derivations And Theory:**

I assessed the sensibility of the derivations and theory.

**Review Assessment: Checking Correctness Of Experiments:**

I carefully checked the experiments.

**Review Assessment: Thoroughness In Paper Reading:**

I read the paper at least twice and used my best judgement in assessing the paper.

---

> ### Author Response · Authors · 2019-11-15
> **Thank you for the comment**
>
> Dear reviewer,
> Thank you for your comment.
> We have indeed started analyzing more complicated cases of nonlinear networks and other architectures, and discovered that the results indeed generalize beyond the linear, fully connected networks.
> 1. NGD (natural gradient descent) perfectly smoothes out the learning dynamics of input-output map making it independent of jacobian, and 2 partial NGD (sqrt-NGD) has the equivalent effect but on the parameter dynamics. The insight from the simple system indeed helps understanding these results for the complex, but they are outside the scope of this paper.

---

### Official Review · AnonReviewer1 · 2019-10-25
**Official Blind Review #1**

**Rating:** 6

**Review:**

Authors analyse curvature corrected optimization methods in the context of deep learning. They build their analysis on Saxe et.al.s work. They show that curvature corrected methods preserve properties of SGD. They also show the disadvantages of layer restricted approximations. They show the importance of time scales in optimization. The paper looks to deep learning from a dynamical systems perspective and hence their experiments are fitting to this framework.
I have checked the analysis and it seems solid. However, my only concerns it the use case. I would like to see such methods show difference in real world datasets. I believe this is a big missing part of the paper.


**Experience Assessment:**

I have read many papers in this area.

**Review Assessment: Checking Correctness Of Derivations And Theory:**

I assessed the sensibility of the derivations and theory.

**Review Assessment: Checking Correctness Of Experiments:**

I assessed the sensibility of the experiments.

**Review Assessment: Thoroughness In Paper Reading:**

I read the paper at least twice and used my best judgement in assessing the paper.

---

> ### Author Response · Authors · 2019-11-15
> **Thank you for your comment**
>
> Thank you for your comment on scaling up the analysis.
>
> To numerically confirm the predictions of our theoretical results, our simulation required estimating and inverting the full Hessian of the system (without approximations). Also, the continuous-time dynamics analysis required using very small learning rate for accurate comparison. The computational cost of full Hessian and the small update size were prohibitive to use real world dataset in this work.  We are indeed implementing application of the new proposed method for real world dataset in our coming work!

---

### Official Review · AnonReviewer2 · 2019-11-01
**Official Blind Review #2**

**Rating:** 1

**Review:**

In this manuscript, the authors analyze the dynamics of training deep linear neural networks under a generalized family of natural gradient methods that apply curvature corrections. They first show that the learning trajectory (direction of singular mode dynamics) in natural gradient descent follows the same path as gradient descent, while only accelerating the temporal dynamics along the path. Moreover, the authors show that the learning trajectory in layer-restricted approximations of natural gradient descent can significantly differ from the true natural gradient. Also, the authors proposed a fractional natural gradient that applies partial curvature correction which in addition to faster convergence, neutralizes vanishing/exploding gradient problems.

I vote to reject this paper due to the following major concerns.
1.	The results of the paper are limited to specific deep linear neural networks which are not practical. More specifically, in addition to only covering the deep linear case, the results hold under the following unexplained assumptions: (a) the correlation matrix of the input data is assumed to be whitened; (b) the singular vectors of adjacent weight matrices are well aligned; (c) Section 4, the singular values of all the weight matrices are the same.

2.	The paper lacks a cohesive introduction that includes a literature review on this very rich area of research.

3.	The paper is not clearly written and is missing many details.

I discuss my comments on the manuscript in details next.

•	While the assumption on the input correlation matrix being whitened can be dropped, other assumptions in the paper are unrealistic and are stated without any intuition or explanation (the singular vectors of adjacent weight matrices are well aligned; (c) Section 4, the singular values of all the weight matrices are the same.)

•	The analysis performed in the paper cover the deep linear case and no intuition is provided on how these results can help in understanding the more general non-linear case. By citing [Saxe et al. 2013], the authors claim that the effect of curvature of loss landscape on deep learning dynamics can be well captured by deep linear networks. However, in [Saxe et al. 2013] this was only shown empirically on a specific example. In recent years, we have encountered many results that hold in deep linear networks but fail to hold when including simple non-linearities. For instance, when the target matrix Y is full rank, every local minimum of the studied problem is global in the deep linear case. However, this result does not hold even with the simplest non-linearities.

•	The paper is unpolished and the objectives and contributions are not clearly stated. Moreover, the results are not rigorously stated and proved. Moreover, the algorithms stated and proposed in the paper are directly used without any introduction or explanation (how they work? What update rules do they use?  …)

•	The introduction in the paper is very short and does not include a literature review on this rich area of research. Hence, the problem is not motivated by a gap in the literature and not linked to other results that study the behavior of algorithms used for training deep neural networks.


**Experience Assessment:**

I have read many papers in this area.

**Review Assessment: Checking Correctness Of Derivations And Theory:**

I assessed the sensibility of the derivations and theory.

**Review Assessment: Checking Correctness Of Experiments:**

I assessed the sensibility of the experiments.

**Review Assessment: Thoroughness In Paper Reading:**

I read the paper at least twice and used my best judgement in assessing the paper.

---

> ### Author Response · Authors · 2019-11-15
> **Thank you for the feedback.**
>
> Dear reviewer,
> Thank you for the feedback. We included major updates and clarifications per your comments.
>
> 1. The results of the paper are limited to specific deep linear neural networks which are not practical.  Unrealistic/Unexplained assumptions.
>
> Our work aims to provide a novel contribution to extending our understanding of deep network's learning behavior beyond the usual gradient descent update regime. The focus on linear networks is an important mathematical tool/framework that allows insightful understanding of the complex phenomenon. It is a necessary limitation as a first step, upon which we hope to develop more generalized theory.
>
> All the assumptions are borrowed from other theoretical works in the field, which are now more clearly referenced. They used for deriving the exact analytical formulas, but they not critical for the general qualitative results. we now include numerical results that supports the generality of the predictions.
>
> 1. Lacks a cohesive introduction:
>
> We have extensively improved our introduction section. Especially we explain that many of the theoretical analysis of deep learning depends on the specific example of gradient descent, and raise the challenge whether those theories generalize under a wider spectrum of learning rules.
>
> 2.
>
> algorithms:

---

> > ### Author Response · Authors · 2019-11-15
> > **Part 2-**
> >
> > The numerical simulations simply used direct estimation of  Hessian, which is numerically inverted to estimate natural gradient. The reference for the detailed methods for Hessian estimation of  Hessian is now included.

---

### Author Response · Authors · 2019-11-15
**General Comments**

We thanks all reviewers for their comments. We have extensively updated our manuscript per reviewers’ suggestions. All major updates are marked in blue. The main summary of the updates are:

1. We included extended literature review and detailed motivation in the introduction. We also clarified our contribution and further polished the overall writing as suggested by  the reviewers.

2. We clearly state the assumptions and limitations relevant for the analysis.

3. We then included additional numerical simulation results that shows our prediction is robust and generalizes beyond the assumption that allowed exact theoretical analysis.  (Figure 3 D, Fig SI1, Fig SI2)

---

### Decision · Program_Chairs · 2019-12-19

**Decision:**

Reject

**Comment:**

This paper aims to study the effect of curvature correction techniques on training dynamics. The focus is on understanding how natural gradient based methods affect training dynamics of deep linear networks. The main conclusion of the analysis is that it does not fundamentally affect the path of convergence but rather accelerates convergence. They also show that layer correction techniques alone do not suffice. In the discussion the reviewers raised concerns about extrapolating too much based on linear networks and also lack of a cohesive literature review. One reviewer also mentioned that there is not enough technical detail. These issues were partially addressed in the response. I think the topic of the paper is interesting and timely. However, I concur with Reviewer #2 that there are still lots of missing detail and the connection with the nonlinear case is not clear (however the latter is not strictly necessary in my opinion if the rest of the paper is better written). As a result I think the paper in its current form is not ready for publication.